# Focus of Attention in Coach Instructions for Technique Training in Sports: A Scrutinized Review of Review Studies

**DOI:** 10.3390/jfmk8010007

**Published:** 2023-01-08

**Authors:** Inge Werner, Peter Federolf

**Affiliations:** Department of Sport Science, University of Innsbruck, 6020 Innsbruck, Austria

**Keywords:** focus of attention, movement pattern, movement technique adaptation, focus effect, optimal feedback control theory

## Abstract

Literature reports superior performance when focusing one’s attention during a movement on environmental effects of that movement (external focus, EF) compared to focusing on the moving body (internal focus, IF). Nevertheless, IF instructions still play an important role in the daily practice of coaches, trainers, and therapists. The current review compiles evidence for focus-of-attention concepts on movement form corrections and technique training. Reviews on the topic and selected additional papers addressing the effect of attentional focus on movement form or on kinetic, kinematic or muscle activity data were included. Both EF and IF instructions affect movement form. The reviews revealed that IF instructions seem to be better applicable to direct movement form changes than EF instructions. In contrast, EF instructions better facilitate optimization within the whole-body coordination, often resulting in better performance outcomes not directly linked to movement pattern changes. Several studies discuss focus-of-attention effects in the context of the optimal feedback control theory expanding on the constrained action hypothesis. In summary, EF and IF instructions both affect form and performance of movements, however, their relative efficacy is situation dependent. The often-purported superiority of EF over IF instructions cannot be generalized to all application contexts.

## 1. Introduction

Instruction and feedback on how a movement is performed are important interactions between coaches and athletes and play an important role in the development of highly skilled movement performances. How exactly these instructions are worded might draw athletes’ attention on how their body performs the movement (internal focus of attention, IF, e.g., “jump as high as possible by opening your knee and hip joints as explosively as possible”); or might direct their attention towards an external goal within the environment (external focus of attention, EF, e.g., “jump as high as possible towards the ceiling of the room”) [1]. For more than two decades, the concept of external and internal focus of attention [2] has attracted growing research interest. For various types of movements, it was reported that EF versus IF instructions result in different movement performances. Some studies were concerned with basic skills like jumping, motor balance or strength exercises [3,4,5], while others examined specific sports skills, e.g., in golfing, basketball, swimming or surfing [6,7,8,9]. Despite the diversity in task demands within these studies (e.g., muscle power, outcome accuracy or all-out performance measurements), there seems to be consistent evidence for EF instructions being more beneficial for motor learning as well as for performance outcomes [10,11] than IF instruction.

The observed difference in movement performance outcomes due to attentional focus has been explained by the constrained action hypothesis [12], saying that concentrating one’s attention on the effect of the movement within the environment (EF) supports automatized motor control and correction circuits. In contrast, focusing attention on the body movements (IF) leads to conscious movement control mechanisms interfering with automatized motor command and movement correction circuitry. This interference leads to constrained movement execution, resulting in less optimal movement results. A corresponding system-dynamics perspective on differences between EF and IF instructions would be that the IF instructions interfere with the mechanisms of self-organization within the motor control systems [13,14]. Hossner and Ehrlenspiel [15] observed that the constrained action phenomenon is only present at points within a movement sequence that are directly addressed by the IF instruction, whereas other parts within the same movement were unaffected. They termed this phenomenon the nodal point hypothesis [15].

However, in contrast to what experimental results and theoretical considerations suggest, IF still plays an important role in the daily practice of coaches, athletes, and movement therapists. In some application domains, IF is even the dominant form of instruction and feedback [16,17,18]. Furthermore, athletes themselves also seem to use internal cues more often [17,19].

The number of studies investigating focus of attention has continuously increased over the last years, as has the number of systematic reviews and meta-analyses on the topic. While in 2013 Wulf reviewed a set of 53 studies, in 2021 Chua and colleagues provided a meta-analysis of 143 studies [10,11]. Within the reviewed original studies, jumping tasks are the most frequently investigated movement task (EF/IF effect on jumping height or distance) [11]. However, measuring height and distance defining movement outcome does not give any information about the changes of the movement pattern or movement form that, in turn, led to the achievement of superior results in EF. In technique training, coaches modulate important parts of movement patterns to achieve a desired improvement in movement form to reach higher levels of performance. Therefore, the impact of focus instructions on movement pattern adaptation is of essential interest and should be considered in more detail.

The current paper intends to appraise evidence of focus-of-attention concepts in movement form corrections to support or decline a paradigm change to EF usage within sport-technical and therapeutic teaching. Thereto it critically assesses and summarizes results of systematic reviews and selected additional recent studies to:identify topic areas for which evidence regarding focus of attention effects is provided and discussed in the available scientific literature;specifically consider EF and IF instructions towards movement form and movement technique corrections.

The discussion of these points is intended to provide a theoretical framework beyond the constrained action hypothesis for the focus of attention effects.

## 2. Materials and Methods

The current study is a narrative review of the up-to-date knowledge on focus instructions applicable to physiotherapy or movement technique training. PubMed, PSYNDEXplus and Web of Science were searched for reviews using the terms “focus of attention”, “motor skill”, or “motor performance”. Only papers written in English were considered. Reviews published since 2011, thus representing the last 10 years of research, were scrutinized. To assess the magnitude of differences, effects sizes were compiled from systematic reviews and are presented here with Cohen’s d (ranging from 0.2 for small effects to 0.5 for medium effects and 0.8 and more for large effects), or with Hedge’s g (with the same ranges). Supplemental articles were added if they addressed the effect of attentional focus on movement form or movement technique and reported on kinetic, kinematic or muscle activity differences. Some articles included in the reviewed reviews were selected for further discussion if they helped clarify specific focus effects.

## 3. Results

Thirteen reviews were found, summarizing more than 250 studies investigating the focus of attention effects in instructions (see Table 1). Five distinct topics areas emerged from the review of these studies for which focus-of-attention effects could be summarized: the effects on motor balance, on explosive skills (e.g., jumping), applications in youth sport or in therapeutic settings, and—being the central question of the current article—results explicitly concerning movement technique.

Motor balance: In particular balance tasks seem to benefit from EF conditions [20], providing small to medium effect sizes for the difference EF versus IF in acquisition and retention and high effect size in transfer tasks (Cohen’s d = 0.48, 0.44 and 1.41, respectively, [21]). EF instructions outperformed IF instructions in achieving higher balance scores. Focus effects were determined by altered body sway (pathlength of center of pressure) or balance-board tilting, being itself the result of complex movement co-ordination within task constraints (movement outcome).

Explosive skills: Similar to the benefit of EF in balance tasks, performance output of explosive jumps and muscular maximum strength increased when EF conditions were applied [1,22]. EF instructions enhanced muscle endurance, especially in lower-body strength exercises [23]. Neumann gathered 16 articles for a systematic review concerning focus effects on weightlifting techniques [24]. EF led to higher peak torque values, accompanied by lower muscle activity measured by surface electromyography (EMG). Thus, muscle commands seem to be more effective in EF conditions for achieving the given task goals. Diversity of designing EF instructions was discussed embracing visual cues like focusing the moving bar or auditory cues like focusing the machine sound, compared to addressing instructions to movement technique and form, which was in any case defined as IF [24]. Chua et al. summarized 15 studies concerning EMG measurements, showing a large effect of EF compared to IF (inducing lower EMG values for EF, Hedge’s g = 0.83), confirming that concentrating on the moving body increases muscle activation when reaching the focused movement goal [11].

Youth sport: Concerning youth sport, two reviews indicate inconsistent results. Barillas et al. reported 35 studies with contradictory outcomes, concluding with preference of EF instructions while respecting person-specific responses [25]. Another systematic review [26] looked for evidence of the OPTIMAL theory for motor learning by Wulf and Lewthwaite [27] and found 35 studies concerning attentional focus strategies. About one third of those studies did not show superiority of EF conditions. Authors explained this inconsistency by developmental differences in young participants, but also by differences in instruction length and complexity, desired direction, addressed movement point, and the dissimilar use of visual cues or metaphors.

Training with patients: Consideration of focused instructions on physio-therapy and training with patients was summarized by four reviews, two of them criticizing insufficient evidence for an endorsement of EF [28,29]. However, newer protocols [30] confirmed the superiority of EF and therefore recommended an EF for recovery exercises. Piccoli et al. [31] reported that superiority of EF instructions could only be confirmed for patients with musculoskeletal disorders and not for patients suffering central nervous system deficiency.

Motor coordination and skill technique: Out of 258 studies gathered in the different reviews, only 43 addressed changes in movement form or movement patterns due to focused instructions. Effect sizes compiled in the extensive review of Chua and colleagues [11] on motor coordination and skill technique turned out to be small (Hedge’s g = 0.26).
jfmk-08-00007-t001_Table 1Table 1Reviews compiling attentional focus studies since 2011.AuthorsYearSystem Rev.Nr. of Studies Observed PopulationKind of TasksOver all ResultBarillas et al. [25]2021No35athletes, youthagility sprinting jumping etc.inconsistentChua et al. [11]2021Yes143all kindsall kindsEF ^1^ > IF ^2^, CON ^3^Grgic et al. [22]2021Yes10young adults:mostly trainedresistance exercisesEF > IF, no long term effectGrgic et al. [23]2022Yes5young adults mostly trainedmuscular enduranceEF > IF, IF = CONHunt et al. [30]2017Yes6patients, older adultsrecovery exercisesEF beneficialKakabeeke et al. [29]2013Yes20healthy and clinicalbalance, dart throw, golf etc. inconsistentKim et al. [21]2017Yes16young adultsbalanceEF > IFMakaruk et al. [1]2020Yes14young adultsjumpingEF > IF, CONNeumann [24]2019Yes16young adultsweightliftingIF > EF in muscle activity EF for competitionPark et al. [20]2018No18mostly young adultsbalanceEF > IF, CONPiccoli et al. [31]2018Yes13patientsbalance dart throw single leg jump etc.inconsistent EF > IF for MD ^4^
not for CNS ^5^ patientsSimpson et al. [26]2021Yes35 to focuschildren, some in young adultssoccer, ball manipulation throwing and jumping balance etc.EF > IF, CON in 21 of 35 studiesSturmberg et al. [28]2013Yes7patients with musculoskeletal dysfunctionbalance, gait relaxation resistance exerciseinsufficient evidence^1^ external focus, ^2^ internal focus, ^3^ control situation without focus, ^4^ muscular dysfunction, ^5^ central nervous system disorder.

## 4. Discussion

Taken together, the results corroborate the way instructions are worded and thus the way attention is focused on aspects of the movement does affect movement performance and does modify movement execution. A large proportion of the scientific literature indicates advantages in EF instructions over IF instructions with regard to movement outcome and learning. However, a closer examination specifically at adaptations in movement-coordination reveals some uncertainty about the expected effects. To classify focus effects on movement technique in more detail, three main questions will be discussed. First, what is the impact on movement form when using EF or IF? Second, how can we apply this knowledge for instructing technique training? And third, what kind of explanation and motor control model could fit to observed behaviors?

### 4.1. Focus Impact on Movement Form

Comparing the effect of instructions used for jumps, typically like “push the floor behind” (EF) versus “open your knee angle explosively” (IF), led to higher jump length or jump height in EF condition [32,33]. Results revealed that IF instruction improved knee angle opening specifically, but at the same time lessened jumping scores. In contrast, the external focus provoked rapid opening of ankle, knee, and hip angles together, leading to an overall improved task performance [34]. Interestingly, peak force (ground reaction forces measured by a force plate) and peak power in a jump-and-reach task did not change, albeit higher jumping performance in EF condition for trained athletes [35]. This result might be due to an optimized co-ordination of all body parts or due to better harmonized timing of movement elements to realize higher jumps.

Comparable observations were made in continuous movement patterns, like running. Movement technique measures in sprinting revealed that expert athletes showed no difference in sprint time or in push-off forces when using IF or EF, whereas collegiate athletes produced slower sprint times with IF, but again without differences in the push-off forces [36]. In another experiment [37] both the IF as well as the EF instructions addressed leg movement in landing and the push-off phase in sprinting. Both focus conditions led to slower sprint times while increasing the vertical component of the ground reaction forces with no change in horizontal direction [37]. Despite a significant change in the movement technique execution according to the instruction, the effect did not result in better sprint times. In the same manner, investigating the swimming start techniques with using the EF (push the blocks away) and IF (push with your feet) instructions, caused no difference in most of the measured variables despite a lower horizontal acceleration in IF condition [38]. Together, these two studies demonstrate the sensitivity of focus consequences, where IF instructions modulate the addressed movement part, but at the same time the whole neuromuscular system is unable to optimize these adaptations for the movement outcome.

Superiority of EF instructions due to enhanced performance outcome is commonly attributed to the assumption of executing better suiting movement patterns. In contrast, studies show inconsistent results. There is evidence in the literature that specific instructions (EF as well as IF) induce movement changes exactly at the addressed movement part. For example, Wulf et al. [39] implemented a dual task in balancing, controlling the horizontal position of a handheld pole while standing on a balance disk. If EF instructions directed attention to the disk position, then the postural sway decreased. In contrast, if the pole position was addressed, the pole movement decreased. Raisbeck and Yamada [40] compared the effect of EF instructions either on jumping height or on landing mechanics when doing drop jumps. Landing mechanics were best when being addressed without subsequently reaching maximum jump height. Jump height was best when addressed in instructions, revealing that landing mechanics in these drop jumps were then comparable to baseline measurements [40]. This means, only addressing jump height with an external focus would not correct drop jump landing mechanics. Generalizing this result, we can hypothesize that EF conditions do not facilitate essential movement trajectory changes but optimize learned co-ordination patterns. Related to that observation, it has to be pointed out that movement form corrections (e.g., knee alignment control in landing maneuvers for injury prevention) are an essential goal in technique training. Thereto, the benefit of an exclusive use of EF instructions is questionable. In consequence, the small effect size in the meta-analysis calculated by Chua et al. [11] for EF over IF instruction on movement technique modifications comes as little surprise.

### 4.2. Application to Technique Training

The intense search for studies investigating movement form when using EF or IF instructions showed that only a few studies on this specific topic have been conducted so far. This mirrors critical differences in the measurement complexity of motor skills observed in focus studies. The relative sparsity of such studies might have two distinct reasons. First, performance outcome measures like jumping height or distance are well defined and straightforward to assess, in contrast to movement technique analysis, which requires rather complex procedures and the selection of representative kinematic and kinetic parameter subsets. Second, whereas performance outcome can be easily addressed with an EF instruction, technique training often requires instructions aimed at the motion of specific body parts, provoking the difficulty to directly address these parts by EF instructions. Some authors automatically define the naming of body parts in the wording as IF instructions [24]. In order to utilize potential positive effects of EF instructions also in technique training, special constructs have to be created that target specific body segment trajectories through environmental effects. Haines et al. [41] used two vertical poles in front of the landing position to address knee positioning in landing and jumping with “pointing your knees towards the poles” (EF) versus “keep your knees over the toes” (IF). In EF conditions knee valgus position at initial contact was lower than in IF condition, while the amount of knee flexion and ground reaction forces did not change [41]. Results of another study [42], where only imaging was used instead of visual control by spatial targets, revealed no difference of IF (keep the middle of your knee in line with the middle of your foot) and EF (keep the knee marker in line with your foot marker). Nevertheless, both focus conditions lead to a significant reduction in valgus angles in landing [42]. Visual help of limb positioning is supposed to support task achievement better than imagery, simplifying the connection of proprioceptive information and task goal achievement [43].

Within that context, Kons [44] could show that the targeted body-range addressed by instructions is of great importance. Constructing an unspecific internal focus (“focus on body extension”) versus a classic IF (“focus on knee extension”) compared with an EF (“focus to reach the ceiling”) revealed that the unspecific IF achieved comparable results to the EF condition. Usually IF instructions are restricted to a small part of the body, whereas EF instructions allow the whole body to act towards the task goal. Therefore, in order to address coordination of movements within the whole body, the use of metaphors is often seen and successful—but should not be mixed up with EF instructions.

Singh and Wulf [45] conducted an experiment working with an image representing the acting body part. They could show that imagining moving a platform (built by the plane of arm position and shoulder axis), lying outside of the body, instead of focusing on the arm-shoulder movement itself, led to more successful volleyball passes [45]. This supports the assumption that if the focus addresses a movement task (like ‘move the imagined platform’), all body parts take part in the whole-body co-ordination related to the desired platform positioning and therefore providing more success. The compelling information whether the IF on the desired arm positioning led to a more accurate one, without fine-tuning to the whole-body movement execution, is not available and should be studied in further experiments. Instructing the movement of limbs directly (move arms and shoulders in a plane), instructing the imagination of moving parts of the body (move like a platform) or imagination of environmental effects (move an outlying platform) show a continuum of instructions providing interaction of the athlete and the environment linked with the task goal. Gose and Abraham [46] underpinned that IF or EF instructing should not be seen as binary choice, but rather as a continuum of options for creating helpful movement descriptions and tasks.

### 4.3. Focus of Attention Mechanisms Interpreted with Reference to the Optimal Feedback Control Theory

Lohse et al. [47] observed changes in success and limb coordination when throwing darts while focusing on arm movement (IF) in contrast to focusing on the dart trajectory (EF). When arm movement was focused, the variability of trial to trail execution was lowered with the consequence of goal accuracy impairment. EF instructions showed an increase in task goal achievement while the variability of trial-to-trial arm movement increased, allowing for compensation of variable moving body parts reaching lower variability in the movement outcome; this is higher accuracy on the dartboard. Convincingly propounded by Lohse and colleagues [47], this phenomenon can be explained by the Optimal Feedback Control Theory [48], predicting that movement variability is only controlled if the task goal achievement is at risk. Instructions to movement execution as a primary task goal shifts movement variability control to the targeted task goal and therefore, defines the control strategy within the movement execution.

Observing muscle activity during motor performance confirmed that for IF conditions, higher EMG amplitudes occur [11,24]. Focusing on the moving body provoked significantly higher muscle activation to secure goal achievement. This phenomenon turned out to be rather body part specific, showing no effect on lower body muscles when focused instructions aim at lifting movements of the upper body [49], as predicted by the nodal point hypothesis [15]. To control movement outcome in EF condition, the co-activation of muscles appears to be downregulated over all to secure an effective muscle play [5] with lowering energy expenditure. Within the Optimal Feedback Control Theory [48] the state estimator—where feedback is used to predict deviations from movement planning—is active to control focused movement success. Feedback of task relevant body positions and movement trajectories and the comparisons between the state estimator prediction and actual movement state help to better achieve the desired movement goal [50]. If attention is directed to only a specific part of the movement (IF condition) then these systems might optimize the addressed part at the expense of the integration of this part into the whole movement execution.

The study of Coker [33] specifically investigated the integration of movement parts into the whole movement. Three different instructions for maximizing a standing long jump were provided in the study: focus on the knee extension (IF), focus on a rapid arm swing (IF2), or focus to reach near the cone in front of the participants (EF). Results revealed that focusing on the arm swing led to longer jumps than focusing on knee extension, while reaching longest jumps in the EF or control conditions. Looking at the projection angles during take-off, arm swing focus provided better projection angles than knee extension focus. This means, when focusing on arm movements, leg extension was optimized to the general task goal (jump distance). While the arm swing itself was hypothesized to significantly change, it was actually less effective within the whole-body movement pattern, resulting in a shorter jump distance than in EF conditions. It is important to note that in this study, jumping distances did not differ between EF and no instruction [33]. Unfortunately, the altered coordination of arm movements was not explicitly observed.

The impact of the frequency of instructions might be interesting information for practitioners. Most of the studies provided instructions immediately before each movement execution or block of trials with reminders to make sure that attention is adequately directed [37,40,45]. None of the reviews referred to the frequency of instructions and potential effects. Instructions have to be clear, short, use precise phrasing, and be adapted to the recipient’s (athlete, patient) understanding [10,20,25,26].

### 4.4. Summary

Summing up the main findings of the reviews and important focus-of-attention literature addressing movement form and technique training, we can state:Even though the body of literature referring to the focus effect is large, only a relatively small number of studies observed movement form corrections and technique adaptation, showing divergent results. IF and EF instructions differed in their impact on movement form and movement technique. However, this effect turned out to be small. Therefore, a general superiority of EF instructions in technique training is not sufficiently legitimate.Literature indicates that EF instructions facilitate optimization of existing co-ordination patterns within movement execution, whereas IF is able to adjust movement trajectories in a desired direction, thus potentially being more capable of movement technique corrections. Results support the implementation of imagining techniques to adapt EF on partial movement form corrections where possible.Study-results of focus effects seem to be in line with ideas of the Optimal Feedback Control Theory. Instructions directing attention to specific movement sequences or specific joints, or segments, should be expected to act on this segment. However, such a focus might interfere with the integration of movement elements into the whole movement, which can lead to a worse movement outcome. External focused instructions then seem to facilitate optimization of movement pattern integration and, as a result, movement execution.

## 5. Conclusions

The huge body of focus studies clearly demonstrates that shifts in attentional focus and subsequent changed success in movement outcome happens by a small change of wording within the movement instruction. This underpins the importance of sophisticated strategies for creating instructions to direct the focus on athletes by trainers and coaches. Some of the systematic reviews promote a positive EF effect, but this notion does not hold for movement form corrections and technique training, where still only insufficient evidence is available. If coaches head for optimization in the whole-body co-ordination, then EF instructions with imagined movement effects seem more suitable. In contrast, IF instructions should be favored to provoke movement form changes and technique corrections. The use of both implicit focused instructions—to become aware of movement patterns—and explicit focused instructions—to provide optimization of whole-body-movement execution, may lead to more success in movement technique training.

## Data Availability

Not applicable.

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
