# Peer review of "Focus of Attention in Coach Instructions for Technique Training in Sports: A Scrutinized Review of Review Studies"

_jfmk, 2023, doi:10.3390/jfmk8010007_

Round 1
Reviewer 1 Report
Title: Focus of Attention in Coach Instructions for Technique Train-2 ing in Sports: a Scoping Review of Review Studies
This article seems well built and brings evidence of a phenomenon not yet fully understood and that certainly deserves further study.
Some points of revision are provided below
Abstract
- I suggest to expand the abstract ……..with a strong conclusion
Introduction
Page 24-31: needed of specific references (now is a lot autoreferential)
I suggest to explain the internal/external focus from a physiological / neurophysiological point of view
Materials and Methods
Specify the inclusion and exclusion criteria for the review and how studies were grouped for the syntheses") and item 8 about the selection process ("Specify the methods used to decide whether a study met the inclusion criteria of the review, including how many reviewers screened each record and each report retrieved, whether they worked independently, and if applicable, details of automation tools used in the process") has not been properly done or not done. In fact, they are not the only items that the authors have not followed. I could relate several more, but one of the most serious is item 11 about the risk of bias assessment ("Specify the methods used to assess risk of bias in the included studies, including details of the tool(s) used, how many reviewers assessed each study and whether they worked independently, and if applicable, details of automation tools used in the process").
This is a very serious error, but more serious is that the authors in the "author contribution" section ensure that "Methodological quality assessments were conducted by all authors". Firstly, nowhere in the review can I find any allusion to whether the quality of the studies was assessed and how it was assessed. I have only found in the results section that the authors state that "the quality and effect of the selected studies were analyzed using an effect size matrix". I hope the authors understand that this is not acceptable.
Results
Authors should be cautious when analyzing effect sizes (ES). Cohen's d is the appropriate effect size measure if two groups have similar standard deviations and are of the same size. Glass's delta, which uses only the standard deviation of the control group, is an alternative measure if each group has a different standard deviation. Hedges' g, which provides a measure of effect size weighted according to the relative size of each sample, is an alternative where there are different sample sizes. The later is important, if you have different sample sizes then you should use Hedges' g.). I am not sure that all the articles meet the criteria to be evaluated with Cohen's d. If the meta-analysis could not be performed, the authors must justify why not and then, justify the analysis of the ES.
Reviewer 2 Report
The Article offers for readers no new information, because last review Article about the topic were published in 2021.
Reviewer 3 Report
General: Interesting review. Generally indicates instruction with an external focus works somewhat better than internal focus. Points out the importance of task specificity.
Additional aspects that would be interesting -
1. does the frequency of instruction make a difference
2. would repeating the instructions often during trials (during or after a movement) make a difference
How much instruction is too much - Explicit versus implicit
Perhaps these items could be addressed very briefly in one or two paragraphs
Specifics: line
36 remove of; Despite the diversity
lines 80 -90 - what languages did you use for the study search
Discussion
line 146 change "se' to indicate: literature indicate ----
line 147 - change look to examination :a closer examination----
Round 2
Reviewer 1 Report
The mani document was well improved
Reviewer 2 Report
Authors don´t accept my recommandations, because don´t include any new relevant information for readers, because last review Article about the topic were published in 2021. From this reason I strongly recommend reject this Article.